# Barriers and Facilitators to Cervical Screening among Migrant Women of African Origin: A Qualitative Study in Finland

**DOI:** 10.3390/ijerph17207473

**Published:** 2020-10-14

**Authors:** Esther E. Idehen, Anna-Maija Pietilä, Mari Kangasniemi

**Affiliations:** 1Institute of Public Health and Clinical Nutrition, Faculty of Health Sciences, School of Medicine, University of Eastern Finland, Yliopistonranta 1, P.O. Box 1627, 70211 Kuopio, Finland; 2Department of Nursing Sciences, Faculty of Health Sciences, University of Eastern Finland, Yliopistonranta 1, P.O. Box 1627, 70211 Kuopio, Finland; annamaijapietila50@gmail.com; 3Department of Nursing Sciences, Faculty of Medicine, University of Turku, 20014 Turku, Finland; mari.kangasniemi@utu.fi

**Keywords:** cervical cancer screening, disparities, disease prevention, health inequalities, health promotion, healthcare service access, migrant health, women health, public health, qualitative study

## Abstract

Globally, cervical cancer constitutes a substantial public health concern. Evidence recommends regular cervical cancer screening (CCS) for early detection of “precancerous lesions.”Understanding the factors influencing screening participation among various groups is imperative for improving screening protocols and coverage. This study aimed to explore barriers and facilitators to CCS participation in women of Nigerian, Ghanaian, Cameroonian, and Kenyan origin in Finland. We utilized a qualitative design and conducted eight focus group discussions (FGDs) in English, with women aged 27–45 years (*n* = 30). The FGDs were tape-recorded, transcribed verbatim, and analyzed utilizing the inductive content analysis approach. The main barriers to CCS participation included limited language proficiency, lack of screening awareness, misunderstanding of screening’s purpose, and miscomprehension of the CCS results. Facilitators were free-of-charge screening, reproductive health services utilization, and women’s understanding of CCS’s importance for early detection of cervical cancer. In conclusion, among women, the main barriers to CCS participation were language difficulties and lack of screening information. Enhancing screening participation amongst these migrant populations would benefit from appropriate information about the CCS. Those women with limited language skills and not utilizing reproductive health services need more attention from healthcare authorities about screening importance. Culturally tailored screening intervention programs might also be helpful.

## 1. Introduction

Globally, cervical cancer (CC) is a commonly diagnosed disease and the leading cause of cancer death amongst women; thus, it is a significant global public health concern [1,2,3]. Data from 2018 indicated that nearly 90% of deaths resulting from the disease were reported from low-and-middle-income-countries (LMICs) [1,3], probably due to limited screening access and the influences of several “social determinants of health” in the LMICs [4,5]. However, evidence suggests that CC is a preventable disease through regular Papanicolaou (Pap) testing [5,6,7]. Hence, cervical cancer screening (CCS) is recommended for all eligible, “at-risk” women for early detection of “precancerous lesions” as a way of reducing new cases and deaths from the disease [6,8]. Nevertheless, women’s active participation and adherence to the screening recommendations are required to achieve these goals. 

The increasing proportion of female migrants globally, with many residing in Europe [9,10], as well as the health status of these population groups [11] calls for more efforts to strengthen healthcare systems [11,12,13,14,15]. In recent years, Finland has witnessed an influx of individuals with migrant backgrounds, including women [16]. For instance, according to a recent Finland’s statistics report (2019), there were approximately 7% of persons with foreign backgrounds in the general Finnish population, representing 21% increase of these populations as compared to the previous year [16]. Furthermore, it has been predicted that these proportions will double by 2030 due to the persistent shortage of workers in Finland, with some estimates that 20,000 more foreign citizens are needed to keep the nation’s economy functional [17].

In Finland, the optimal aim of healthcare services is “to maintain and improve people’s health, well-being, work, functional capacity, social security, and reducing health inequalities” to all legal inhabitants and their families, irrespective of their cultural background and socioeconomic conditions [18]. Finland provides free mass CCS at five-year intervals to all residents, including legal migrants, mainly in the age range 30–60 years. Those individuals identified from the Finnish Population Register receive a personal screening invitation letter [19]. Presently, there is about 70% participation in the mass screening program, although the aim is to achieve 80–85% screening coverage [19]. Additionally, opportunistic testing (unorganized) is widespread in the country [20]. Even where a universal free healthcare service exists, as is the case for CCS, it does not automatically ensure utilization of such services, suggesting that there may be impediments hindering access to the services [21,22].

Previous studies in Finland about CCS in Finland, which were based on a quantitative survey, and register data demonstrated disparities and lower cervical screening participation among women with a nonnative mother tongue [23,24,25], i.e., in some migrant populations [26,27] as compared with the general Finnish population (Finns). Migrants are a heterogeneous population, and their socioeconomic status can vary depending on their educational level, occupation, and reason for migration. These features might influence their access and utilization of healthcare services [28], such as CCS. There seem to be disparities in CCS and a lower likelihood of screening participation among several migrant origins, even in some countries with screening opportunities [26,27,29,30,31,32,33,34,35]. Both a higher incidence and risk of CC have been revealed among some migrant populations compared with the host population [22,36,37,38]. Ultimately, these persistent health inequities will increase national healthcare expenditure [39].

Several individual-related barriers to CCS participation have been identified, e.g., unmarried, unemployed, illiteracy/lower education level, and language difficulties [27,31,33,35,37,40]; women’s lack of knowledge/awareness of cancer/risk factors or screening practices, unpleasant experiences of screening such as pain, embarrassment, and obesity [34,41,42,43,44,45,46]; and advanced age and no history of pregnancy or birth [27,30,33,47,48]. Other factors have been considered, such as migration-related issues in the host country [31,33,35,49], women’s fear of cancer or test results, and cultural and religious beliefs [43,45,50,51,52,53,54]. System-related barriers include mistrust of healthcare authorities, inaccessibility to healthcare or/and interpretation services, and concern about the gender of the test-taker [40,45,54]. However, gaps in knowledge exist about qualitative studies investigating the factors influencing CCS participation among migrant origin in countries with screening opportunities. It is imperative to identify the factors hindering screening participation among various migrant-origin groups if we are to enhance the screening participation within these populations and prevent a significant incidence of CC and mortality resulting from disease [8].

Therefore, this study aimed to explore barriers and facilitators to CCS (Pap) testing participation in women of Nigerian, Ghanaian, Cameroonian, and Kenyan origin in Finland. Our research questions were as follows: (i) How do women understand CC and CCS (Pap) testing practices? (ii) How do women perceive the facilitators to CCS participation in Finland? (iii) What do women perceive as the barriers facing them in accessing and participating in CCS in Finland? (iv) How do these women propose that the screening protocol be improved for migrant populations in Finland?

## 2. Materials and Methods

### 2.1. Study Design

Our study used a qualitative design. In general, this technique is appropriate for exploring individuals’ views, experiences, perceptions, ideas, feelings, and motivation on particular matters [55,56]. We employed focus group discussions (FGDs) for data collection. FGDs offer an effective way of studying phenomena that can be discussed in groups and of using group dynamics to generate new data [55], which would not be possible to obtain with a quantitative method. This study employed the Standards for Reporting Qualitative Research Reporting (SRQR) [57] as a guide for reporting this study.

### 2.2. Study Settings

This study was carried out in two migrant Christian organizations and a university campus in Helsinki and eastern regions of Finland. Upon obtaining ethical clearance from the University of Eastern Finland Ethics Committee (#14/2015), the researcher (E.E.I.), a member of some of the communities under study, identified the women and two migrants’ Christian organizations. These women and the two church facilities were purposely selected to reach a reasonable number of women from different African communities and capture their diversities [51].

These two church premises usually accommodate, on average, from 50 to 150 (adult men, women, and children) from different migrant backgrounds. These are mainly Africans who typically assemble in these church premises mostly on Sundays to worship. Later, the researcher (E.E.I.) attended and presented the study background to the two migrants’ Christian organizations’ leaders to seek permission for recruiting participants. Engaging the Christian organization’s settings and involving members of the participants’ ethnic backgrounds are considered viable ways to recruit and reach a representative sample of these populations and to gain their trust despite their diverse ethnicities [58,59]. Women from the university campus from eastern regions of Finland and the same countries of origin were approached through word-of-mouth and snowball approaches [60].

### 2.3. Participants and Recruitment

Women of Nigerian, Ghanaian, Cameroonian, and Kenyan origin in Finland were purposely selected [61] and recruited through a convenience sampling approach. The selection criteria for the women were as follows:Women from English-speaking African countries (to decrease the language barrier, for easy abduction of data, and for accuracy without translation).Women age 25–60 (main age range for CCS recommendation in Finland) [19].Women having a residence permit for at least three years in Finland before the study (to ensure their acquaintance with the Finnish healthcare system).Women residing in the country’s capital areas (possibilities of reaching a high proportion of participants, as most of these migrant populations live in these areas) [62].

Additionally, a university campus was chosen from the eastern region of Finland for recruiting participants for the study. Women from the university with the same age range and countries of origin and using the same selection criteria were recruited through word-of-mouth and snowball approaches [60]. Thirty women were enrolled in the study; twenty-five were from the two Christian organizations and five were from the university campus (E.E.I.). After obtaining consent from the women to participate in the study, each woman’s contact information (telephone numbers) was obtained to arrange suitable dates and times for each FGD meeting. Participants were given a choice of conducting the FGDs, like a quiet and familiar environment outside the church premises or in the university, like meeting rooms as a convenient location.

### 2.4. Data Collection/Procedure

Data collection commenced from September 2016 to June 2017. The FGD approach was used to explore perceptions of CC and CCS (Pap) testing practices, barriers, and facilitators to CCS participation in women of Nigerian, Ghanaian, Cameroonian, and Kenyan origin in Finland. Also, the women’s suggestions for improving the CCS protocol for the migrant populations in Finland were explored. The FGDs’ themes were created based on previous studies [42,44,63]. The sheets for background information, FGDs’ themes, and guides were piloted [55] in August 2016 with one focus group of five women. Based on the piloting responses, a few minor changes were made to the text of the themes. The FGD’s guide was constructed into two parts:Background information which the women were asked to fill their responses to assess the following: age, educational level, marital and employment status, country of origin, years, municipality of residence in Finland, religion, level of Finnish/Swedish (Finland’s two official languages) skills, and number of children. We also assessed women’s history of Pap tests taken in Finland by asking women the following questions: if they have ever undergone a Pap test in Finland in the past 1–5 years (yes/no/when). If yes, they were asked how/where they received the information about the screening program in Finland.The FGDs’ topic guide was focused on the following topics: individual perceptions of CC and CCS (Pap) testing, perceptions of barriers and facilitators to CCS participation in Finland, and the women’s suggestions for improving the CCS protocol for the migrant population in Finland.

The researcher (E.E.I.) conducted, guided, and moderated all the FGDs. Each FGD session began by welcoming and thanking the women for turning up to the FGDs. To introduce the participants to the group discussions, they were given information about the study and a short presentation about CCS, known as the Pap test, and about the CCS procedure in Finland, i.e., women were informed that CCS in Finland are tests offered to women from 25–65 or 30–60 years of age, through invitation letters sent every five years, and the test is free-of-charge [19]. Some pamphlets about the screening and its purposes were presented to the women, and they were requested to fill in the background information sheet.

Altogether thirty women participated in the study. We had seven FGDs, consisting of three to five participants in FGDs and one pair discussion. Small focus groups were used to achieve a comfortable shared discussion on the research topic’s sensitive nature as the size was easy to manage [55,61]. The FGDS were held in English separately with women recruited from the two migrant Christian organizations and a familiar quiet environment in the church and two separate FGDs in the university’s meeting rooms to ensure confidentiality [61]. With the participants’ consent, all discussions were audiotaped [55]. The average duration of each FGD was about 45 min–1 h. The pair discussion lasted for about 45 min; thus, approximately six hours of recordings were generated.

### 2.5. Data Analysis

We analyzed the data by utilizing the inductive content analysis approach [64] to provide sensitive knowledge about participants’ perceptions about CCS participation in Finland. This approach is considered appropriate in qualitative studies, particularly when exploring individual perceptions about a phenomenon under study [64,65] and where limited knowledge exists about the phenomenon under investigation [64]. Thus, based on the inductive content analysis process, increased knowledge of the research subject can be generated [65].

In the first phase, the discussions were transcribed verbatim and anonymized [55,65]; this generated 83 pages of text. The analysis started by reading the data carefully several times to achieve familiarity and immersion in the data [57] to correspond with the study’s aims. Subsequently, the analysis unit, word, or couple of words or sentences of the content were coded, grouped, and categorized into subthemes [64,65]. Next, the categories expressing the main contents of the data were analyzed and coded to identify central themes (E.E.I. and M.K.). The use of colors and cut-up papers were assigned to meanings of different categories to make sense of the participants’ comments linked with quotes (E.E.I., M.K., and A-M.P.). Next, the subthemes were abstracted in the main themes and named inductively based on the data [64]. The analysis was checked when it was initially conducted and rechecked by all authors; all agreed with the results.

### 2.6. Ethical Considerations

The University of Eastern Finland Ethics Committee approved this study (#14/2015). Additionally, each participant signed a written informed consent form prior to attending the FGD meeting. The women were all informed about the confidential nature of the collected data and the possibility of withdrawing their participation throughout all stages of the research work, as drafted in the Declaration of Helsinki Ethical Principles and the Finnish National Board on Research Integrity (TENK) [66,67]. The women did not receive reimbursement for study participation; however, women with young children were provided childcare during the FGD session. To ensure confidentiality, women’s personal information was not gathered; they were allocated with a number assigned during the meetings.

## 3. Results

### 3.1. Participants’ Characteristics

Altogether, 30 women participated in the study, with over half (*n* = 19) reported having participated regularly in CCS in Finland. Screening participation from each municipality was Helsinki (*n* = 10), Espoo (*n* = 3), Vantaa (*n* = 3), and eastern regions of Finland (*n* = 3). The women’s age ranges were 25–45 years, the mean age was 30 years, and all were Christians. The women had higher education levels, e.g., the majority had a university education (*n* = 22); most were married or cohabiting (*n* = 20), and most women had 1–2 children (*n* = 17). A few women were unemployed (*n* = 5); many were students or caring for their children at home (*n* = 17). They had lived in Finland for one to twelve years, with over half having resided in the country from one to five years (*n* = 18) (Table 1).

Four broad themes emerged from the FGDs and pair discussions analysis, which were intertwined at the individual and screening system levels as individual perceptions of CC, causes, screening (Pap testing), barriers to CCS participation, facilitators to CCS participation, and participants’ suggestions for improving the CCS protocol (Figure 1).

The results are presented based on our participants’ perceptions of the above themes. The subthemes identified from the themes are described, illustrated with quotes from the participants, and displayed below.

### 3.2. Theme 1: Perceptions of Cervical Cancer and Screening

Our participants had varying levels of knowledge of CC and its causes, and CCS (Pap) testing practices, i.e., ranging from considerable understanding to no understanding. Women described their understanding of CC as a common disease among women, unfamiliarity with CC, and fear of cancer.

#### 3.2.1. Understanding of Cervical Cancer

##### A Common Disease Found among Women

Among the study participants, only a few women expressed a significant understanding of CC. They described CC as a common disease among women globally.

##### Unfamiliarity with Cervical Cancer

A majority of the women stated that they were unfamiliar with the disease. In contrast, they stated they were only aware of breast cancer and described the disease as one of the major health issues among women.

##### Fear of Cancer

Some of the women described CC as a fearful and dangerous disease carrying a death sentence. Others described the disease based on their cultural perceptions about cancer in general.

#### 3.2.2. Understanding of Causes of Cervical Cancer

##### Human Papilloma Virus/Sexual Behavior

Some women knew and described the Human Papilloma Virus (HPV) as a leading cause of CC, stating that the risk was related to sexual behaviors. Meaning that CC can be transmitted sexually or through a woman having several sexual partners.

##### Multiple Births

A few of the women ascribed the risk of CC to multiparity. That is, they meant that a woman who has had numerous births could be at risk of developing the disease.

##### Miscellaneous Causes

Women had misconceptions about the risk factors of CC. For example, some believed a higher risk of the disease among older women, having had a family history of cancer, and other causes such as consuming certain foods or a woman utilizing tampons for extended periods. Others stated that they had no understanding of the causes of CC.

#### 3.2.3. Understanding of Cervical Screening

##### Health Promotion Intervention for Women

Many of the women described CCS as a “health promotion intervention for women”; thus, they described the screening as a means for promoting and raising awareness of female health. They also viewed screening as a willingness to improve their health and to have the knowledge and participating in screening allowed them to take care of their health.

##### Cervical Screening not Understood

Many women expressed a misunderstanding of the screening (Pap test), meaning that they had not heard about it or had not participated in the screening in either their country of origin before migrating to Finland or in Finland. The themes of participants’ statements and quotes are summarized below (Table 2).

### 3.3. Theme 2: Barriers to Cervical Screening Participation

The barriers raised by the women were influenced by the levels of the individual and screening system. These comprised six subthemes: (i) Individual-level barriers included limited language proficiency, communication issues, and unpleasant experiences of screening, and (ii) system-level barriers were lack of screening information, misunderstanding of screening’s purpose, and miscomprehension of screening results.

#### 3.3.1. Individual-Level Barriers

##### Limited Language Proficiency

One recurring issue expressed by most women was that, since their proficiency in Finland’s official languages (Finnish/Swedish) was inadequate, this represented a significant barrier to their participation in the screening. They explained the difficulties in reading and understanding the information in the screening invitation, and they needed someone to translate its contents to them or else they had to call the healthcare center for further explanations.

##### Communication Issues

Some participants mentioned communication issues related to their guidance during screening procedures by the healthcare professionals performing the test. A minority mentioned that they had encountered communication problems, such as having received no or minimal information about the screening procedures. Similarly, cultural/race-related issues were raised, and they believed that these experiences might be related to their skin color.

##### Unpleasant Experiences of Screening

A minority of the women stated that they had experienced some physical pain and received unfriendly attitudes from healthcare professionals during the screening procedure. These women thought such experiences could affect their future screening participation.

#### 3.3.2. System-Level Barriers

##### Lack of Screening Awareness

As described by many unscreened women, the main barrier to screening participation was their complete unawareness of the CCS program’s existence in Finland. These women further described that they had not received any screening invitations despite their extended stay and even have had children born in Finland. The women stated that they had become aware of the screening only during their participation in these FGDs.

##### Misunderstanding of Screening’s Purpose

Several screened women stated that, although they were aware of the Finnish CCS procedure in Finland and knew that the test was called the “Pap test” and, therefore, they participated, most felt they had misunderstood the screening’s purpose.

##### Miscomprehension of Screening Results

Many of the screened women explained that, although they had participated in the test, they had miscomprehension of screening results, i.e., stating that they did not receive test results and were unaware of the next step after taking the test. Consequently, they assumed that not receiving the test result could mean that everything was “normal” and stated that there was no need to expect a test result. A few women mentioned that they had called the health center or booked an appointment to obtain their test results (Table 3).

### 3.4. Theme 3: Facilitators to Cervical Screening Participation

The women’s perceptions of facilitators to screening were influenced by the individual and screening system levels and comprised of the following five subthemes: (i) the importance of screening and health professionals and (ii) the system facilitators were reproductive health services utilization, free-of-charge screening services, and source of screening information in Finland.

#### 3.4.1. Individual-Level Facilitators

##### Importance of Screening

Some of our participants expressed that their primary motivation for screening participation was their understanding of CCS‘s significance for early detection of CC. The women further explained that, if the screening recommendations were not adhered to and the disease was not detected early, it could impact an individual’s health, even being fatal.

##### Health Professionals

Some women appreciated the test taker’s expertise and stated that it had been a “pleasant experience”. They described that they had received detailed guidance and understandable information and were treated with compassion and kindness throughout the screening procedure.

#### 3.4.2. System-Level Facilitators

##### Reproductive Health Services Utilization

Most screened women shared a common view that their primary source of access or awareness about the CCS program in Finland was their contact or referral from reproductive healthcare clinics. Thus, these women expressed the belief that it was only mothers who had access to CCS services in Finland.

##### Free-of-Charge Screening Services

Many screened women viewed the free access to the screening services available in Finland as their primary motivation in the screening participation.

##### Source of Screening Information in Finland

As indicated by a few screened women, another principal facilitator to CCS was related to their awareness of accessibility through the screening invitation letters they received from their residence municipality (via the organized mass screening program system). A minority mentioned that they had undergone opportunist (non-organized) tests arranged by their gynecologist and screening information obtained through relatives and friends, who had encouraged their participation (Table 4).

### 3.5. Theme 4: Participants’ Suggestions for Improving Cervical Screening Protocol

Based on the identified barriers, the women expressed their views about how they believed the screening protocol could be improved for increasing participation in migrant communities. These included a more diverse language approach for resolving the language problem, disseminating screening information more widely, and enhancing health professionals’ competencies.

#### 3.5.1. Approaches to Cervical Screening

##### Language Approach

A common view among the women was that one crucial way of improving screening participation would be to provide appropriate information in screening invitations and test results in a language that they can understand. They mentioned that the most straightforward way to avoid language problems would be that the screening invitation and test results could be written in more languages, e.g., Finnish and English.

##### Appropriate Information about Screening

Several women further suggested that more details could be given about the screening/results. They emphasized the need to supplement the screening invitation with a short explanation of what the screening entailed in English. This would allow women to have a clearer understanding of possible follow-up procedures.

#### 3.5.2. Dissemination of Screening Information

##### Creating of Screening Awareness

Almost all the women agreed that screening information should be disseminated through migrant organizations to become aware of the screening program. Other suggestions included social/mass media, schools, and healthcare centers.

##### Individuals with Similar Cultural Backgrounds

The women wished that the screening authorities would employ individuals with the same cultural or diverse experiences to disseminate screening information. They thought that this would improve screening awareness and help migrants understand the importance of screening since that would facilitate their participation.

##### Using Reminder Letters/Telephone/Recall System

Women felt that sending reminder letters about the screening would probably motivate them to participate in the screening if possible and to follow up with a phone call.

##### Enhancing Health Professionals’ Communication and Cultural Skills

The women viewed improvements in health professionals’ communication and cultural skills as good ways to facilitate their participation in screening. This meant that they would appreciate having a nurse who can communicate in a friendly way with a caring attitude during the screening procedure, making it much more likely that they would return for the next screening appointment. Another suggestion was that it would be advantageous to be attended by nurses who had some cultural knowledge of other countries, especially countries in Africa (Table 5).

## 4. Discussion

This exploratory qualitative study explored cervical cancer knowledge, screening practices, barriers, and facilitators to CCS participation in women of Nigerian, Ghanaian, Cameroonian, and Kenyan origin in Finland. There is a paucity of qualitative studies about the factors influencing CCS participation among migrants to countries with screening opportunities. For the first time, a qualitative study about CCS among these migrant groups has been conducted in Finland to our best knowledge. This study has provided novel insights and new knowledge and enhances our understanding of what determines migrants’ women participation in CCS. Ultimately, these findings can help policymakers develop the screening protocol to improve screening participation in migrant-origin populations. In this regard, our work contributes to the existing knowledge about screening practices in migrant communities worldwide.

Our results indicate that women’s participation in screening was either facilitated or impeded by several factors that emerged at individual and screening system levels, consistent with earlier international qualitative findings among various migrant groups [40,42,44,45,63,68], especially women of African origin [51,52,53,54,69]. Many women in our study expressed relatively poor knowledge about CC and its causes; in accordance with previous studies, they revealed misconceptions about CCS practices [42,43,44,45,51]. This might probably be related to the lack of similar schemes in the women’s country of origin or/and unawareness of screening in the host country [41]. However, most women in our study had heard about the Finnish screening program and were aware of its importance and had participated in screening. Having a sound understanding of health information and what it entails seemed to raise the women’s motivation to utilize the CCS service; health literacy and healthcare service utilization may thus be related [70,71].

The total unawareness of the existence of the screening in Finland, as evidenced by the numbers of unscreened women, i.e., those stating that they had never received an invitation to CCS despite their extended stay in the country and having given birth to children in Finland, agrees with earlier studies [41,43,44]. Therefore, it is imperative that women be aware of the screening program and understand the information in the invitation letters and the follow-up procedure. This information should explain the rationale for the screening; consequently, this might motivate their participation.

The most commonly mentioned barriers were related to language difficulties and misunderstanding of the invitation letters’ contents, again consistent with previous findings [33,40,45,52,54,63]. In Finland, at the time of this study, the invitation letter is written in both official Finnish languages (Finnish/Swedish); this explains the women’s difficulties in reading and understanding the screening information. Similarly, this language/communication problem might also be responsible for those who reported having participated in the screening in Finland but felt that they had misunderstood the screening’s purpose and were unaware of the screening results, as reported in a previous study [43]. Also, a few women admitted they might have received a screening invitation letter/result, but due to the language difficulties, they might have assumed that it was an advertisement and thrown away the letter. Moreover, the surprising result was about women’s expressions that they had not obtained test results after taking the test and assumed that not receiving the screening results automatically implied that the test result was “normal”. Thus, the importance of proper language knowledge in screening participation cannot be overemphasized and has also been demonstrated in a previous quantitative population-based study among some migrant-origin individuals in Finland [27]; thus, it is crucial that there should be an understanding of the contents of invitation letters if we are to achieve effective communication between the receiver of care and its supplier.

As reported previously, only a minority had encountered a degree of unpleasantness or physical pain during screening [40,42]; however, a woman’s screening experience is crucial in determining the likelihood that she will return for the next appointment [43,44], and this was expressed by some of the women in our study. In contrast to earlier findings [49,51,54,69], although all the women in our study were Christians and from different cultural backgrounds, issues related to religious background and ethnicity were not particularly prominent factors to screening participation; instead, most women had positive attitudes towards screening.

The awareness of the free-of-charge screening services offered in Finland as a motivator to screening participation agrees with an earlier study [42]. This corresponds with the World Health Organization‘s “goal of universal healthcare access and that all people can obtain the health services they need without financial suffering” [72]. The right of access to universal healthcare services includes screening, like CCS to eligible women. However, persistent health inequities due to healthcare inaccessibility or/and financial unaffordability ultimately increase national healthcare expenditure [39].

Our result demonstrates that the utilization of female reproductive healthcare services as an access route to screening supports a previous finding from Finland [27] and international reports [47,48]. In fact, since most of the women in our study were of reproductive age, they were more likely to have been provided with all kinds of health information in addition to screening information during antenatal care visits.

Furthermore, the limitations of the screening invitation from their residence municipalities as a common source of screening awareness or facilitator for screening participation were often mentioned. In Finland, the primary method for scheduling mass CCS is through a personal invitation letter, which is usually valid for a year [19]. Similarly, physician’s recommendations or visits to a gynecologist were mentioned as ways to inform about screening, in agreement with earlier reports [42,73], as opportunistic testing is also widely used in Finland [20].

The barriers raised by the majority of the women and the fact they frequently suggested that the information in the screening invitations should be written in a language they understand (English) accords with earlier international studies [45,54]. In our study, these women comprehend English since it is the official language in their countries of origin. As suggested by the women in our study, other investigators have also stressed the need to strengthen the health system by providing appropriate health information [14,15], to create screening awareness through migrant organizations, and to utilize a telephone recall system, which might encourage their participation in screening [45,54].

### Trustworthiness

This study’s credibility was guaranteed as the researchers had different experiences with qualitative investigations and one of the researchers was familiar with the migrants. Participants were recruited from different ethnic groups; in these situations, the FGD is a suitable data collection method since it gathers reliable data.

We recognized that data saturation was achieved after five FGDs due to the repetition of participants’ responses of perceptions of CC, CCS, barriers, and facilitators to the screening participation [74,75,76], as the preliminary data from the FGDs were simultaneously transcribed verbatim. However, we decided to conduct three more FGDs, including the pair interview, to verify that we had truly reached data saturation. Furthermore, the researcher kept the tape recordings and took field notes during the FGD meetings for backup to enable rechecking of the participants’ responses for accuracy. All researchers rechecked the transcripts and analysis to verify the accuracy of the participants’ comments. The study’s confirmability/transferability was ensured by describing the study’s protocol, i.e., selection criteria, settings, and data collection/procedure. The participants’ responses from the transcripts have been described and illustrated with relevant quotes. The study dependability was assured by using the same discussion guide as the FDGs conducted by the same moderator, who was an “insider” of some of the communities being studied [77], thus increasing data consistency. All FGDs and pair discussions were held in a language with which the participants were familiar (English) with no need for translation.

## 5. Limitations and Strengths

This study has some limitations. Firstly, being an “insider” and based on one’s experience may result in a difficulty in “separating” oneself from the group under study, which might affect the data analysis [77,78]. Secondly, there might be bias relating to the selection criteria, as women from non-English speaking countries were excluded from the study. Thirdly, although we aimed to carry out FGDs with a maximum of five women in each group, to stimulate a free discussion on the topic’s sensitive nature, five women approached for the study declined to participate but agreed with previous reports [55,59]. Hence, there were smaller numbers of participants in some FGDs, e.g., one group had only two women. Thus, as a limitation, a sharing atmosphere was not achieved in the discussions held in some groups. However, these discussions’ contents were included in the data to respect the participants’ willingness to provide information on this sensitive subject. Fourthly, women’s responses were based on self-reporting, posing the possibility of recall bias [79], as women might not recall if they have received the screening invitation letter. Lastly, a qualitative study aims to obtain a deeper understanding of the topic under investigation and is not intended to be generalized; for example, these results cannot be extrapolated to all African migrant communities living in Finland due to the study’s small sample size.

This study’s key strengths are that the study applied the qualitative research design and FGDs, regarded as an appropriate way of exploring the migrant women’s understanding, experiences, and perceptions of barriers and facilitators related to CCS participation. Furthermore, data were collected by a woman researcher with a nursing background, experience in both qualitative and migrant studies, and an “insider” of some of the participants’ communities [77,78]. Thus, we believe that reliable data was obtained, especially about this kind of sensitive information that might not be available to an “outsider” [78,80]; this assumption was confirmed by two of the participants: “*If this discussion/information is given by ‘white people’, many would not want to come. If she (moderator) is a ‘white lady’ talking, maybe I will not flow the way I am flowing now. As she is giving the information, you can see, we are just flowing. If she is a white lady (moderator), she would not even break it down the way we would understand or want” (P1/3/FGD6).* The other researchers have a nursing background and extensive experience in conducting qualitative studies. Prospective studies on the current topic with larger groups are recommended to identify other barriers to screening participation among the migrant populations.

## 6. Conclusions

Though Finland provides free-of-charge mass CCS to all residents, including legal migrants, our results indicate that women’s participation in the screening was either facilitated or impeded by several factors that emerged at the individual and the screening system levels. The main barriers to CCS participation among women were language difficulties and lack of screening information. Enhancing screening participation amongst these migrant populations would benefit from appropriate information about the CCS. Those women with limited language skills and not utilizing reproductive health services would need more attention from healthcare authorities to inform them about the importance of screening. The simplest ways to resolve these problems would be to write the screening invitation and test results in English/other languages and to disseminate detailed information about screening through migrants’ organizations and the mass/social media. Culturally tailored screening intervention programs and improving healthcare professionals’ communication and cultural competencies might also be helpful.

## Figures and Tables

**Figure 1 ijerph-17-07473-f001:**
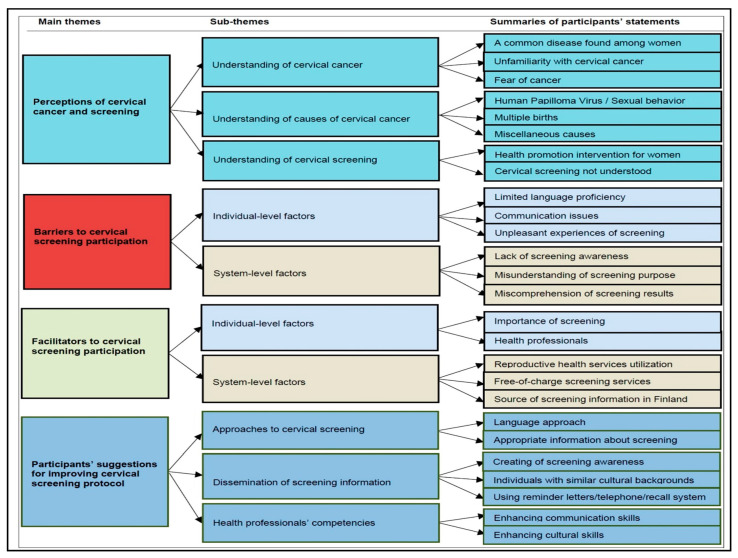
Summary of the main themes and subthemes emerging from the group discussions.

**Table 1 ijerph-17-07473-t001:** Characteristics of study participants.

Total Participants (*n* = 30)	All Groups	Group	Group	Group	Group	Group	Group	Group	Group
Age (Years) (25–45)		1	2	3	4	5	6	7	8
Mean Age (30)		(*n* = 4)	(*n* = 4)	(*n* = 3)	(*n* = 4)	(*n* = 5)	(*n* = 3)	(*n* = 2)	(*n* = 5)
Description	Total n	n	n	n	n	n	n	n	n
Country of origin									
Nigeria	17	2	1	1	3	3	3	1	3
Ghana	5	1	0	2	1	0	0	1	0
Cameroon	5	0	2	0	0	2	0	0	1
Kenya	3	1	1	0	0	0	0	0	1
Educational level ^1^									
University degree	22	4	3	3	3	5	0	2	2
High school	7	0	1	0	1	0	3	0	2
Basic school	1	0	0	0		0	0	0	1
Marital status									
Married/cohabiting	20	4	2	3	1	4	2	1	3
Single/divorced	10	0	2	0	3	1	1	1	2
Employment status									
Employed	8	1	1	1	2	0	1	0	2
Unemployed	5	2	0	0	0	1	2	0	0
Student/childcare	17	1	3	2	2	4	0	2	3
Religion									
Christian	30	4	4	3	3	5	3	2	5
Length of stay in Finland (years)									
1–5	18	3	2	2	1	5	2	1	2
6–12	12	1	2	1	3	0	1	1	3
Literacy in Finnish/Swedish									
Moderate	11	1	2	2	2	1	1	0	2
Fair	13	2	0	1	1	3	2	2	2
Poor/not at all	6	1	2	0	1	1	0	0	1
Number of children per woman									
1–2	17	2	1	3	2	4	0	1	4
More than 2	4	2	1	0	0	1	0	0	0
Poor/not at all	6	1	2	0	1	1	0	0	1
None	9	0	2	0	2	0	3	1	1
Have had a Pap test in Finland									
Yes	19	3	2	3	2	4	2	0	3
No/do not know	11	1	2	0	2	1	1	2	2
Source of screening information in Finland									
Invitation letter ^2^	8	1	1	1	1	0	2	0	2
Reproductive health service	11	3	1	0	2	3	0	0	2
Opportunist test	5	0	0	1	1	1	1	0	1
Friends/relatives	2	0	1	1	0	0	0	0	0

^1^ Education obtained in any country. ^2^ Mass screening invitations by municipalities (Helsinki, Espoo, Vantaa, and eastern regions of Finland).

**Table 2 ijerph-17-07473-t002:** Examples of quotations of perceptions of cervical cancer and screening.

Understanding of Cervical Cancer
Common Disease Found among Women “I know cervical cancer is the kind of cancer that affects the cervix, and it is mostly found in women.” (P4/FGD5) “Well, my understanding of cervical cancer is the cancer of the cervix.” (P1/FGD7)
Unfamiliarity with Cervical Cancer
*“The type of cancer that I am familiar with over the years is breast cancer that is like a major problem all over the world.” (P1/FGD1)* *“I do not know anything about cervical cancer, and I do not know what causes it. I have no idea.” (P2/FGD6)*
Fear of Cancer
*“I think cervical cancer is cancer that affects a lady or the female cervical part, and it is dangerous to have it. It is a deadly issue that has to be taken seriously.” (P1/FGD8)* *Cancer in our cultures is taboo; many people do not talk about it openly, especially if you are a woman from Africa, it is a death sentence.” (P4/FGD2)*
Understanding of Causes of Cervical Cancer
Human Papilloma Virus/Sexual Behavior *“The leading causes, as I know, is the HPV (**Human Papilloma Virus) and multiple sexual partners.”**(P1/FGD3)* “It is cancer, and if you have sex with somebody, it can be transferred.” (P4/FGD5)
Multiple Births
*“Having too many children/multiple births and having them consecutively, then you have a higher risk of it (CC).” (P1/FGD2)* *“I think that can also cause, like, giving birth, getting pregnant, and all that.” (P5/FGD5)*
Miscellaneous Causes
*“Most people that have these things (CC) are older people.” (P2/FGD4)* *“I do not know much since it can come from our daily activities, food intake, and genetics.” (P1/FGD6)*
Understanding of Cervical Screening
Health Promotion Intervention for Women “I think cervical screening is one measure or intervention for women to reduce cervical cancer incidence like the Pap test.” (P1/FGD3) “I think that it is important, promoting female health, it is for my good and, that is motivating enough.” (P3/FGD2) Cervical Screening not Understood “I have never taken a Pap test in my life, where I came from, nobody has ever told me to go for it.” (P2/FGD3) “I have no information on how they do the screening.” (P2/FGD7)

**Table 3 ijerph-17-07473-t003:** Examples of quotations of barriers to cervical cancer screening (CCS) participation.

Individual-Level Barriers
Limited Language Proficiency “I am motivated to go, but the language is a barrier. If the letter comes in Finnish, I will not read it, and I will throw it away because I do not understand.” (P3/FGD3) “I think many people would not go to the screening because they do not understand what the letter is about when they get the letter. (P2/FGD8)” Communication Issues “In my case, she (test-taker) did not even explain the test. Only she asked if I have done the test before or not.” (P2/FGD4) “If you are looking at me differently because of my skin color and treat me like that, I will not be likely to encourage my fellow sisters to go (screening).” (P4/FGD5) Unpleasant Experiences of Screening “My own was just the pains, the physical pains when she (test-taker) was searching; it was painful.” (P3/FGD5) “I think that even the experience can affect the screening because if you have someone nice, you will feel motivated to come back.” (P2/FGD1)
System-Level Barriers
Lack of Screening Awareness “I have been here (Finland) for like seven years, and given birth to two children, I have always had a permanent address and never got that kind of letter.” (P3/FGD8) “The awareness is so limited; I did not even know what we were talking about until now.” (P4/FGD2) Misunderstanding of Screening Purpose “I did not know that the Pap test is for cervical cancer, so there was no fear of anything. I was not thinking about how good the test is to me at that moment, but I went because everybody was going. I was thinking I will be punished for not going.” (P3/FGD7) “The letter comes home, it is now left for you to know what is written, it is the same process for everybody.” (P1/FGD2) Miscomprehension of Screening Results “I thought after the test that I would be told about the results. Until now, I have not gotten results, and do not know what the results were.” (P4/FGD5) I think sometimes they do not tell you the result if you are clean. If they see a problem, then they will call you immediately to get your treatment.” (P2/FGD1)

**Table 4 ijerph-17-07473-t004:** Examples of quotations of facilitators to CCS participation.

Individual-Level Facilitators
Importance of Screening “I knew a friend who had cervical cancer. If she had this awareness and gone for the test, maybe she would have still been alive today, and because she did not do the test (Pap test), she eventually died.” (P4/FGD5) “I would like to do it without fear, and when they do the checking and find that something is going wrong, they will give you some cure for it, not allowing it to advance.” (P4/FGD2) Health Professionals “The experiences were good; the nurses spoke good English, and they made everything comfortable.” (P2/FGD7) “The approach I received during the first test was good. I was made to understand and clearly explained what I was going through, that was enough motivation for me to have it again.” (P1/FGD6)
System-Level Facilitators
Reproductive Health Services Utilization “It was after I had given birth, a midwife just told me to go for a Pap test. She described what the process was and gave me an appointment.” (P4/FGD8) “Until I had my baby, although I have been living here (in Finland), I did not know how it could be made known to those who are not having children. Because we all have children, so we were able to know. So, if a woman just came (recently migrated), and she is not having any more children, how will she be aware of this (CCS)?” (P4/FGD1) Free-of Charge Screening Services “I was overwhelmed by the fact that this could be done, and it was for free.” (P2/FGD8) “I was excited to receive something good from this check-up, in Africa, it would cost so much money.” (P1/FGD7) Source of Screening Information in Finland “I have gotten the invitations twice to come for the Pap Smear.” (P1/FGD3) “I think you need to go to the gynecologist or some doctor and get it checked there or something like that. For me, when you go for a gynecological check-up, they start with a Pap test.” (P3/FGD6)

**Table 5 ijerph-17-07473-t005:** Examples of suggestions for improving screening protocol.

Approaches to Cervical Screening
Language Approach “To put the screening information in different languages they can understand, for most Africans, it is English, that would help.” (P2/FGD8) “The language, there should be better awareness. We are immigrants; our native language is not Finnish, and it is not everyone that speaks Finnish very well.” (P1/FGD1) Appropriate Information about Screening “I think the invitation letter and result can be given in simple English language with a simple and short explanation of what the test is about, and what would be done if anything is found.” (P2/FGD1) “It is about my health; it is good to understand the test as much as I want the care.” (P2/FGD6)
Dissemination of Screening Information
Creating of Screening Awareness “One way the municipality could help get the information to immigrants is to get in touch with some group or community leaders, where immigrants are gathering.” (P1/FGD4) “As immigrants, we have places where we meet, and there we might understand things better than in a letter that comes to our home.” (P3/FGD1)
Individuals with Similar Cultural Backgrounds “If they can involve people with a similar cultural background, it is easier to comprehend such information.” (P1/FGD3) “If they get someone who understands the immigrants, or if they can involve not necessarily someone from Africa, even a Finn, who understands these people and has the time to talk to them.” (P3/FGD1)
Using Reminder Letters/Telephone/Recall System “It might help if they (immigrants) do not show up, send another letter, and follow-up on people and call.” (P1/FGD2) “I can imagine if I get a few letters, and then a phone call... that, you need to make an appointment, because sometimes people forget.” (P3/FGD1)
Enhancing Health Professionals’ Communication and Cultural skills
*“I think if they should put people who are culturally sensitive and understand Africa issues, or wherever this woman is coming from so that when the person asks a question, they can understand.” (P4/FGD1)* *“Also, to address the fear thing. They (nurses) should make people know that having cervical cancer is not a death sentence and that there are measures available if something is being detected.” (P1/FGD8)*

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
