# Peer review of "Barriers and Facilitators to Cervical Screening among Migrant Women of African Origin: A Qualitative Study in Finland"

_ijerph, 2020, doi:10.3390/ijerph17207473_

Round 1

Reviewer 1 Report

The article is well written and the study is good with rich data. There are a few areas to be reconsidered so that the article is in better shape for publication. The coding and presentation of results were done properly.

  1. Justification. Why is the study timely and how is it justified? Why knowing about migrants in Finland important?
  2. Research questions. The research questions are more relevant to a quantitative study. A question like “What are the barriers...?” refers rather to identifying opinions or in this case facilitators and barriers rather than exploring participants’ understanding. So the question should be phrased differently, such as “How participants understand the barriers?”.
  3. Methodology. I am not sure about the use of the “inductive content analysis” to analyse participants’ expressed experiences. Content analysis is largely used for analysing written documents. If it is also used for understanding people’s expressed experiences then this has to be better documented. Otherwise, the “General Inductive Approach” as presented by Thomas (2006) is relevant. The Intepretivist Phenomenological Approach is also pertinent because as the study is about participants’ experiences.
  4. Data robustness. Data saturation has not been mentioned in the study. Data saturation has been debated by some scholars but others have highlighted its importance in qualitative research. It is included in the SRQR standards you are using. In any case, the question still remains and needs to be answered: How do you know that you have enough data to answer your research questions? You may use Constantinou et al’s (2017) paper about the importance of saturation and its link with qualitative research quality markers and Guest et al’s (2016) paper “How many focus groups are enough?”

Reviewer 2 Report

This is a well-written article. They authors provided enough background information to support their work, explained their methods were and why they chose those method, properly identified themes with supporting quotes and the discussion section was also written well.

However, the only concern I have is the size of participants in some of the focus group discussion was very low for focus group discussions. For example in group 3 the size of group was 3 participants, group 6 three participants and group 7 two participants. These numbers are low for focus group discussions. The ideal size of focus group discussion is between 5 and 8. With only 2 participants, it is no longer a focus group but more or less face to face interview. Maybe this could be acknowledged as a limitation.

Minor comment. The author stated "For the first time, a qualitative study about CCS among these migrant groups has not been conducted in Finland to our best knowledge." I think they wanted to say "For the first time, a qualitative study about CCS among these migrant groups has been conducted in Finland to our best knowledge"

Reviewer 3 Report

Thank you for this well designed, executed and written study. There a few places where the English was a little awkward. I've marked a few of them but overall, it seemed appropriate.

Reviewer 4 Report

The overall manuscript was well written and the topic was interesting. The authors provided sufficient background information pertaining to the topic, selected an appropriate study design, provided a clear explanation of the implemented methods, and was able to present a conclusion that was supported by the results. However, the presentation of the example quotes and themes starting on line 205 should be considered for revision. I would suggest inserting the quotes and the correlating themes and sub-themes into a table for a clearer presentation. 

Round 2

Reviewer 1 Report

Thank for considering my comments and suggestions!